# Improving fake news classification using dependency grammar

**Kitti Nagy**[1]*, **Jozef Kapusta**[1,2]

**1** Department of Informatics, Constantine the Philosopher University in Nitra, Nitra, Slovakia, **2** Institute of Computer Science, Pedagogical University of Cracow, Kraków, Poland

* kitti.nagy@ukf.sk

**Data Availability Statement:** All relevant data are within the manuscript and its Supporting Information files.

**Funding:** This work was supported by the Slovak Research and Development Agency under the contract no. APVV-18-0473. The funders had no role in study design, data collection and analysis,

## Abstract

Fake news is a complex problem that leads to different approaches used to identify them. In our paper, we focus on identifying fake news using its content. The used dataset containing fake and real news was pre-processed using syntactic analysis. Dependency grammar methods were used for the sentences of the dataset and based on them the importance of each word within the sentence was determined. This information about the importance of words in sentences was utilized to create the input vectors for classifications. The paper aims to find out whether it is possible to use the dependency grammar to improve the classification of fake news. We compared these methods with the TfIdf method. The results show that it is possible to use the dependency grammar information with acceptable accuracy for the classification of fake news. An important finding is that the dependency grammar can improve existing techniques. We have improved the traditional TfIdf technique in our experiment.

## 1. Introduction

Fake News is currently a huge problem across all of the world. By expanding social media, the amount of new information is spreading rapidly [1]. There is no universal definition for fake news. Hiramath and Deshpande [2] specify the word fake news as misdirection, gossip, fraud and deception. They state that false information has a big influence on the world. Zhang et al. [1] present fake news as misinformation or hoaxes spreading through both traditional print news media and recent online social media. The internet makes it easy for humans to communicate, not only through social media, but also via email, or other web pages. Alongside increasing e-communication, the amount of false news, hoaxes and other hate speeches is also growing [3]. Nowadays, social media is even more used for expressing people's own opinions, giving them the ability to connect with other people, who share the same viewpoint. Their points of views do not have to be true, but the excessive sharing of the same information makes people tend to believe it and think it may be right [4]. This is a huge problem. Accordingly, this problem of the easy spreading of right and false news, has made its correction even harder but, on the other hand, very needed and attractive. Researchers are trying to detect false news using a variety of methods, from word-based analysis, through syntactic and semantic analysis to different classification algorithms such as statistical-based, and also using machine learning [3, 5, 6].

decision to publish, or preparation of the manuscript.

**Competing interests:** The authors have declared that no competing interests exist.

Using content analysis, we can identify language patterns and writing styles for true and fake news and capture the most important elements for detecting false information. Most of the fake news creators are using specific writing strategies [7]. "Bag-of-words" and "n-grams" are the most common ways of representing information. However, the simplicity of these approaches leads to deficiency in text processing, for example, the problem in "n-gram" model is the extreme tenuity, and the model "bag-of-words" can lose important information by ignoring the context and semantics of words [8, 9].

The morphological and syntactic analysis seems to improve the methods of analysing the content of texts [10]. Syntax analysis is about determining the importance of words in sentences. We also need to consider grammar rules to define the logical meaning as well as the correctness of the sentences [11]. The syntactic analysis also referred to as syntax analysis or parsing, is the process of analysing natural language with the rules of formal grammar. Grammatical rules are applied to categories and groups of words, not individual words. The syntactic analysis basically assigns a semantic structure to the text and within it, we are looking for the so-called "syntactic dependency". A syntactic dependency is a relation between two words in a sentence with one word being the governor and the other being the dependent of the relation and it often forms a tree [12].

Currently, there are many principles for detecting fake news such as n-grams, syntactic and morphological text analysis, POS tags, TfIdf method. These methods can be used for other applications, such as feature extraction, sentiment analysis, or other classification tasks [13–16]. Dependency grammar is widely used in grammar error detection and correction [17, 18]. Researchers use both statistical and machine learning methods for classification tasks [3, 5, 19]. An existing limiting factor in designing effective models for detecting fake news is the low number of available and suitable text corpora [9].

This paper analyses the available dataset of fake and true news using syntactic dependency. One of the outputs of the syntactic analysis is to determine the importance of the words in terms of their syntax. Within the created syntactic tree, we can determine the depth i.e., the importance of each word. This expresses the importance of terms in sentences.

The verb, which is the most important word of the sentence, has the lowest depth. In our demonstration (Fig 1) a direct dependence on the verb is found for the words–mouse and lion. In this way, a simple sentence structure "mouse awakened lion" was identified. The next step is to find the words related to the words mouse and lion i.e., words from the previous level. This approach can also be used to determine the importance of the words. Words like "awakened", "lion", or "mouse" are more important than "little" and "the strong" in the examined sentence.

We used these dependencies for document vectorization in our research. Vectors, created from documents, are used as input vectors for creation, training, and testing classifiers in natural language processing classification tasks. We will present a proposal of our own methods for the improvement of the traditional TfIdf method in this article. TfIdf is a technique to quantify a word in documents, we generally compute a value to each word which signifies the importance of the word in the document and corpus. Our methods will be based on word dependency. Their contribution is compared in the article with the traditional TfIdf method. The following methodology is used for evaluation of the suitability of the proposed method:

1. Identification of word dependencies in the sentences of the analysed dataset.

2. Determining the importance of the words based on word dependencies.

3. Application of our proposed methods, creation of input vectors for the classifier.

4. Creation of input vector also using the traditional TfIdf method.

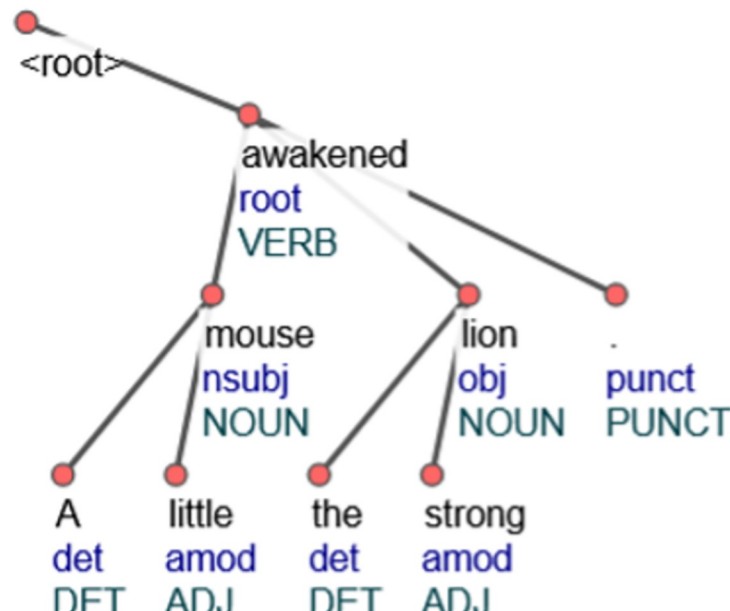

**Fig 1. Graphical representation of the dependency of terms in a sentence.**

5. Design and creation of classification models.

6. Identification and comparison of the performance of the created models, mainly the accuracy.

The paper aims to find out whether it is possible to use the dependency grammar to improve the classification of fake news. Frequently used technique TfIdf does not consider the relationships between words in the sentences. The motivation of our research was to verify whether these relationships can be added to the vectors used for classification. By improving the TfIdf technique, we expect an increase in classification accuracy. We want to compare the created models and verify, whether we can improve classification of news using the dependency grammar in combination with TfIdf method.

The structure of the paper is as follows: The current state of the research in the field of fake news identification is summarised in the second section. The datasets of news used in the research, as well as related pre-processing techniques and the creation of the models are presented. The most important results are summarised in the fourth section. The discussion and conclusions form the content of the last section of the paper.

## 2. Related work

News is an important part of our daily life because they facilitate our understanding of what is happening around us. While in the last centuries it took more days or months to receive news about what happened on the other side of the Earth, nowadays it only takes minutes. However, problems arise as news increases rapidly. Especially by expanding social media, it has become even more difficult to recognise which news is true, and which is false. Therefore, we are forced to look for new types of defence tools.

An in-depth analysis of existing fake news detection approaches and research works can be found in the paper of Hakak et al. [20].

A text pre-processing survey was introduced by Sriyanong et al. [21]. Their work was aimed at text mining on Big Data Infrastructure, but the described principles are variously usable. All data consists of structured and unstructured part, and they must be pre-processed before a computation. It is one of the most important tasks and it includes more steps. They designed and developed an efficient text pre-processing framework on a big data infrastructure, which is proposed to support the text pre-processing task to reduce the computation time. The framework consists of the main four modules: data collection module, a storage module, data cleaning module and feature extraction module.

Kadhim [22] compared feature extraction techniques. He evaluated techniques BM25 and TfIdf. His results show that the performance of TfIdf is better. BM25 ranks a group of documents depending on the query keywords that appear in each document. TfIdf is a usual method for text pre-processing commonly used to feature extraction and its performance is still competitive with new techniques. Due to the success of the TfIdf method, researchers are looking for and proposing various improvements to the method. One of this improvement is a method called TF-MONO presented by Dogan and Uysal [23]. The MONO strategy can use the non-occurrence information of terms more effectively than existing term weighting approaches. Other improvements are findable in other contemporary surveys [16, 24, 25].

Hiramath and Deshpande [2] provided a system for fake news detection, which is based on classifications using Logistic Regression, Naïve Bayes, Support Vector Machines, Random Forest and Deep Neural Network. They achieved the highest accuracy using the Deep Neural Network. Similarly, Kaliyar [19] classified news using Naïve Bayes, Decision Tree, Random forest, K nearest neighbour and LSTM algorithms. He also explored the benefit of the feature extraction using n-gram and TfIdf methods. He achieved the highest accuracy using the Naïve Bayes method.

Hakak et al. [15] extracted important features from the fake news dataset and used them to three popular classification models namely, Decision Tree, Random Forest and Extra Tree Classifier. They achieved a high accuracy for both datasets–Liar and ISOT.

Qawasmeh et al. [26] investigated the automatic identification of fake news over online communication platforms using machine learning techniques. Their model was applied on a dataset with 85.3% accuracy performance. The used dataset consisted of news articles that were labelled as true or false. They assumed that all features in the dataset are important and therefore, they did not apply the cleaning of the dataset and did not remove stop words. The feature extraction was implemented using the Word2Vec approach with Google's pre-trained word embedding model. They tested more models, two of them had satisfying results: Bidirectional LSTM concatenated with accuracy 85% and Multi-head LSTM with the accuracy of about 83% but higher precision.

Kapusta et al. [27] created a morphological analysis of several datasets of news in which they analysed morphological tags and compared the differences in their use in fake and real news articles. They assigned a morphological tag which was deeply analysed for each word. Grammatical classes were created using morphological analysis as a process of classifying the words into grammatical-semantic classes. They found statistically significant differences between fake and real news data mainly in verbs and nouns.

However, we also find approaches based on dependency grammar. Dashtipour et al. [28] provided a work with a novel hybrid framework for concept-level sentiment analysis in the Persian language in which they create concepts from words based on symbolic dependency relation.

Named Entity classification is used in different natural language processing tasks. Grammatical dependencies are very essential to classify a Named Entity. Ahmad et al. [29] used in their research the grammatical dependencies to train a classification LSTM model, which classified the Named Entity and they achieved more than 60% in F1 score and showed an increase in performance of more than 6% on an average. Their experiments also showed that LSTM performed better than other methods for Named Entity classification.

Alothman and Alsalman [17] provided a dependency grammar-based research for an Arabic grammar auditor. They called their initiative developed auditor as "Arabic Grammar Detector". They implemented it based on the dependency grammar and decision tree classifier model. Its purpose is to extract patterns of grammatical rules from a projective dependency graph in order to designate the appropriate syntax dependencies of a sentence. Their detector can detect more than 94% of grammatical errors. Very similar research for English grammar correction based on dependency grammar was proposed by Zhou and Liu [18]. Comparative analysis of related work presented in this paper is shown in Table 1.

**Table 1. Summarization of research studies.**

| Authors | Approach | Model and methods | Dataset | Features, Keywords |
|---|---|---|---|---|
| Sriyanong et al. [21] | Designed and developed a text pre-processing framework on a big data infrastructure | Apache spark | Sentiment data from Facebook page named "Tasty" | Text pre-processing, reducing computation time |
| Kadhim [22] | Comparison between BM25 and TfIdf | BM25, TfIdf | Tweets collection | Feature extraction, term weighting |
| Dogan and Uysal [23] | Novel term weighting approach | SVM, KNN | Reuters-21578, 20-Newsgroups, WebKB | Supervised term weighting |
| Fan and Qin [16] | Improved TfIdf that takes into account the relationships between classes | TfIdf, TF-IDCRF, Naïve Bayes | Corpus provided by Fudan University | text classification |
| Wu and Jia [24] | Improved feature weighting algorithm using chi-square statistical method and category concentration of keywords | Original and improved TfIdf | 10 topics text set from Sogou Laboratory | Feature extraction, text classification, feature weight |
| Long and Yan [25] | Proposed a text feature selection method based on Rank-IDF | Rank-IDF, Textrank, TopicRank, PositionRank | Collection of documents from China Patent Database | Feature selection, TfIdf, text mining |
| Kaliyar [19] | Used natural language processing, machine learning and deep learning techniques to classify fake news | Naïve Bayes, Decision Tree, Random Forest, K-nearest neighbour, LSTM, CNN&LSTM | Various datasets from https://www.kaggle.com | Fake news classification, comparison of different methods |
| Hakak et al. [15] | Extracted important features from datasets and used them in the classification models | Decision Tree, Random Forest, Extra Tree Classifier | Liar, ISOT | Fake news detection, feature extraction |
| Qawasmeh et al. [26] | Proposed an automatic identification of fake news based on machine learning | Bidirectional LSTM, Multi-head LSTM | news articles from Emergent Dataset | Feature extraction, fake news identification |
| Kapusta et al. [27] | Words were classified into classes; grammatical categories were assigned to them and differences between classes of fake and real news were examined | TreeTagger, statistical analysation | Multiple datasets from Kaggle and KaiDMML merged | Text mining, comparison of fake and real news |
| Dashtipour et al. [28] | Proposed a novel hybrid framework for Persian sentiment analysis | CNN, LSTM | Persian product and hotel reviews corpora | Dependency-based classifier, sentiment analysis |
| Ahmad et al. [29] | grammatical dependencies used to learn a classification model which classified the Named Entity | LSTM | DBpedia corpus, Wikipedia corpus | Named Entity classification, dependency grammar |
| Alothman and Alsalman [17] | Arabic grammar auditor was implemented based on dependency grammar and decision tree classifier model | dependency grammar, decision tree | essays from different collections, Tashkeela corpus | Dependency grammar, detection of grammatical errors, suggestions, and possible corrections of the errors |
| Zhou and Liu [18] | English grammar error correction was implemented based on classification model with increased classification accuracy | proposed model based on neural network | not specified 1568 English sentences with grammar errors and corrections | English grammar error correction algorithm |

**Table 2. Dataset description.**

| Dataset | number of all words | number of unique words | average number of words in a record | the shortest record | the longest record |
|---|---|---|---|---|---|
| KaiDMML | 187710 | 11889 | 463.48 | 5 words | 4624 words |
| Covid-19 dataset | 668695 | 17159 | 579.46 | 5 words | 5035 words |

## 3. Materials and methods

### 3.1 Datasets and data pre-processing

We used two datasets in our research. First is a freely available dataset KaiDMML (https://github.com/KaiDMML/FakeNewsNet). This dataset contains fake and real news, and it was originally constructed for the end-to-end system FakeNewsTracker [30, 31]. The dataset consists of verified fake and real news from fact-checking websites. Then, using Twitter's advanced search API, they gather the tweets related to fake and real news that is spread on Twitter. The dataset is multi-dimensional information related to news content, social context and spatio-temporal information and its size is relatively small (208 fake news and 197 real news).

The second used dataset is a Covid-19 dataset. It was analysed in detail by authors (https://towardsdatascience.com/explore-covid-19-infodemic-2d1ceaae2306). This dataset consists of news articles and social network posts on Covid-19 which were labelled true and false. Originally the dataset has 1164 records, we were able to use 1154 records (573 real and 581 fake) after cleaning. A basic statistic of datasets is shown in Table 2.

We cleaned the data and dropped null values in the pre-processing part of our work. We have not removed the stop words, because without them we cannot correctly analyse the text by the dependency grammar. The stop words are in the text generally the most common words, so these are rated as low numbers using TfIdf. Simultaneously, these words are also on the bottom of sentence structures, so dependency grammar assigns them also a low number. Because of these two facts, if we combine two very low numbers, the result will be also a low number and it does not have a big impact in classification. We used the tool UDPipe (https://ufal.mff.cuni.cz/udpipe) for identification datasets dependency grammar. It is a Python prototype, capable of performing tagging, lemmatization, and syntactic text analysis. The prototype first tokenizes the given text, split multi-word tokens into individual words, POS tagging and lemmatization is performed and finally, dependency parsing is accomplished [32].

### 3.2 The techniques used to pre-process and create the input vectors

We are struggling to improve the TfIdf method by using dependency grammar. The used algorithms for classifying fake news are Random Forest and Linear SVC in this paper. The process of our method, classification and evaluation can be seen in Fig 2.

We considered the following techniques: TfIdf, Dgw and MultipleDgw to create the input vector for classification.

**3.2.1 Term Frequency—Inverse Document Frequency.** Term Frequency–Inverse Document Frequency (TfIdf) is a traditional technique that leveraged to assess the importance of tokens to one of the documents in a corpus [33]. The TfIdf approach creates a bias in those frequent terms highly related to a specific domain is typically identified as noise, thus leading to the development of lower term weights because the traditional TfIdf method is not specifically designed to address large news corpus. Typically, the TfIdf weight is composed of two terms: the first computes the normalized Term Frequency (Tf), the second term is the Inverse Document Frequency (Idf).

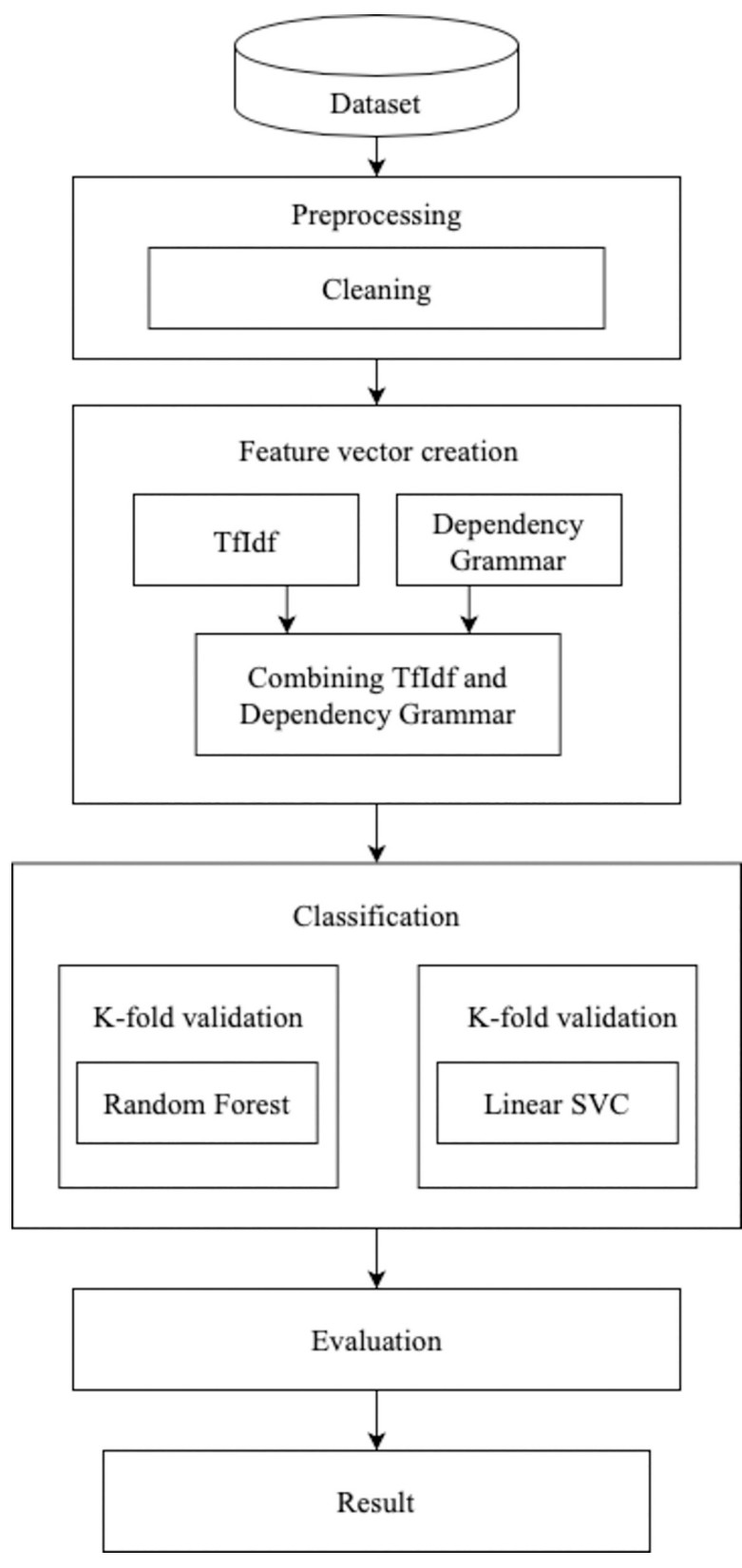

**Fig 2. Workflow of our proposed method.**

Let $t$ be a term/word, $d$ be a document, $w$ be any term in a document, then the frequency of the term $t$ is calculated as in Eq (1):

$$tf(t,d) = \frac{f(t,d)}{f(w,d)} \tag{1}$$

where $f(t, d)$ is the number of terms in the document $d$ and $f(w, d)$ is the number of all terms in the document. When calculating TfIdf, the number of all documents in which the term occurs is also considered. We denote this number as $idf(t, D)$–inverse document frequency and we can express it as in Eq (2):

$$idf(t,D) = \ln \frac{N}{\sum(d \in D : \ t \in d) + 1} \tag{2}$$

where $D$ is the corpus of all used documents and $N$ is the number of documents in the corpus. The formula of TfIdf can be written as in Eq (3):

$$tfidf(t,d,D) = tf(t,d) \times idf(t,D) \tag{3}$$

Formula $tf$ have various variants such as $\log(tf(t, d))$, $\log(tf(t, d)+1)$. Similarly, the $idf$ has several variants of the calculation [34]. We performed the TfIdf calculation using the methods of the scikit-learn library (https://scikit-learn.org) in our experiments, and this method was used as the base method for comparison with the newly proposed methods.

**3.2.2 Dependency grammar weight (Dgw).** Our first proposed technique is Dgw. We identify dependency grammar using the tool (https://ufal.mff.cuni.cz/udpipe). The verb from the sentence is usually identified first and other dependencies are derived from it. We want to emphasize in our approach that verb and derived nouns, appropriately adjectives are more important within the sentence as prepositions, conjunctions or other parts of speech. For this reason, weights are assigned to each word based on dependency grammar. When calculating the weight, the order (depth) of words is the basis. We assign variable *depth = 0* for the so-called "root word" (mainly verb). For words, which were identified as directly depending to root word, we assign variable *depth = 1*, for next-level words we assign *depth = 2*, etc. The maximum *depth* value identified in our KaiDMML dataset is 17 and in the Covid-19 dataset, it is 15.

An existing problem in calculating the weights may be the fact that the analysed text could contain the same word with different depths. The verbs usually have *depth = 0*, but other parts-of-speech could have different depths, which often depend on the number of words in the sentence. However, it should be noted that the difference in the found depths for the same word is not big.

For example, the word "mouse" has in our example in the introduction *depth = 1*. If the same word were to be found in other sentences of the analysed text, its depth will probably be 1, 2, maybe 3. The word "mouse" is a noun, so its depth will not be a large number. For this reason, we calculated the average depth for words that occurred more than once in the analysed records.

Based on these considerations of the depth variable, calculated by the dependency grammar, the weigh vector was designed.

Let $t$ be a term/word, $d$ is a document, $\overline{depth(t,d)}$ is the mean of all variables *depth* for term/word in sentences of document $d$, $c$ is the whole corpus of all documents and *maxdepth (c)* is the identified maximum depth in the whole corpus, then dependency grammar weight

*dgw* for term/word *t* in a document is calculated as follows in Eq (4):

$$dgw(t, d) = \frac{maxdepth(c) - \overline{depth(t, d) + 1}}{maxdepth(c) + 1} \qquad (4)$$

By the above calculation, we will ensure that words with a small depth will have a large weight i.e., weight 1 or near to 1. At the same time, the calculation is derived from min-max normalization and the following formula (5) applies for values *dgw(t, d)*:

$$0 < dgw(t, d) \leq 1 \qquad (5)$$

The vector is calculated using *dgw* for each document. Some vectors have zero values i.e., for terms that are not in the document. We also ensure that *dgw(t,d)≠0* by calculation.

**3.2.3 MultipleDgw.** We use the weights of each word in documents calculated by the proposed Dgw for improving the TfIdf method. We emphasise the value of TfIdf of each word by Dgw weight in the technique MultipleDgw. Let $\overrightarrow{TfIdf(d)}$ be a vector calculated for the document *d* using TfIdf method (6) and $\overrightarrow{Dgw(d)}$ is a vector calculated using Dgw method (7):

$$\overrightarrow{TfIdf(d)} = (t_1, t_2, \ldots, t_n) \qquad (6)$$

$$\overrightarrow{Dgw(d)} = (w_1, w_2, \ldots, w_m) \qquad (7)$$

vector for MultipleDgw is calculated as the multiplication of elements as in Eq (8):

$$\overrightarrow{MultipleDgw(d)} = (t_1 * w_1, t_2 * w_2, \ldots, t_n * w_n) \qquad (8)$$

Where $t_i$ and $w_i$ are values calculated for the same term in the document *d*.

## 3.3 Random Forest classifier

The Random Forest was first introduced by Ho in 1995 [35] and later extensions of the algorithm were developed. It is a supervised machine learning algorithm and it produces great results [3, 36]. The aim of this method is to build multiple trees in randomly selected subspaces of the feature space. Classification of the trees in different subspaces are generalized in complementary ways and combined classification can improve the result. The generalization needs to be made independently and a discriminant function that combines the classification given by the individual trees and preserves their accuracies is also needed. Trees are constructed in randomly selected subspaces of the feature space. Fig 3 demonstrates the architecture of the Random Forest [37]. The algorithm with *N* trees can be described as:

A training data subset from the dataset is created for each of *N* decision trees. This step is also called bootstrap or bagging.

A decision tree is created for each data sample which is trained on one subset. The tree is built until it reaches the maximum size without pruning.

Prediction is made either as voting or average is counted. Mostly so-called Majority Vote is used. We chose this method for its high performance, ability to handle binary and numerical features, high dimensionality and also for quick prediction and good training speed. Default parameters were used.

## 3.4 Linear SVC

Support Vector Machines are a set of supervised learning methods used for regression, detection of outliers and for classification. It can solve linear and non-linear problems. The main

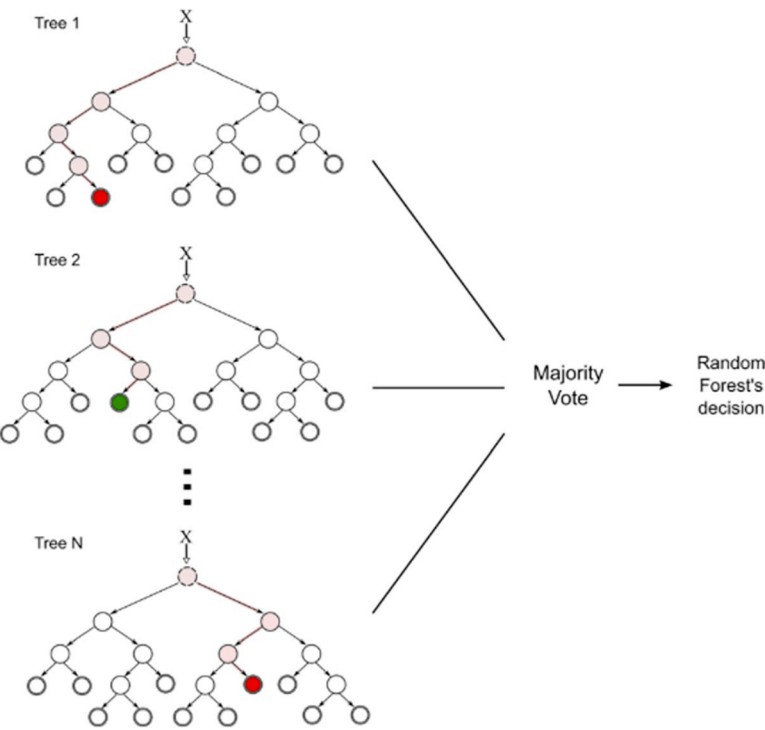

**Fig 3. Architecture of Random Forest [37].**

idea of SVC is to create a line or a hyperplane which separates the data into classes. Linear SVC is one of the SVM methods that can perform binary and also multi-class classification on a dataset.

The purpose of Linear SVC is to fit the data and return the best fit hyperplane that divides the data into two classes. The Linear SVC is similar to the SVC, but the kernel used for classification is linear and it minimizes the squared hinge loss. One-vs-all multiclass reduction is used by Linear SVC. In some cases, it can be faster to compute the result as using similar methods [38].

Linear SVC is used in NLP classification tasks, for example in sentiment analysis problems [39, 40], hate speech detection [5] or other tasks [41].

## 3.5 K-fold validation and evaluation metrics

We used k-fold validation for evaluating our models. This technique is popular and easy to understand. It generally results in a less biased model compared to other methods, because it ensures that every observation from the original dataset has the chance of appearing in the training and test set.

We first shuffle our dataset in k-fold cross-validation, so the order of the inputs and outputs are completely random. We do this step to make sure that our inputs are not biased in any way. The original sample is randomly partitioned into $k$ equal sized subsamples. A single subsample from the $k$ subsamples is retained as the validation data for testing the model, and the remaining $k—1$ subsamples are used as training data. The cross-validation process is then repeated $k$ times, with each of the $k$ subsamples used exactly once as the validation data.

The advantage of this method is that all observations are used for both training and validation, and each observation is used for validation exactly once. We set $k = 10$ in our evaluation, so the dataset was randomly split into 10 subsamples.

The evaluation metrics in our experiments is the classification accuracy. Accuracy is the ratio of correct predictions to the total number of samples and is computed as (9):

$$acc = \frac{TP + TN}{TP + TN + FP + FN} \tag{9}$$

where TP represents the number of True Positive results, FP represents the number of False Positive results, TN represents the number of True Negative results, and FN represents the number of False Negative results. To evaluate, analyse and describe the results of our model we also calculated precision (10), recall (11) and F1 score (12) as follows:

$$precision = \frac{TP}{TP + FP} \tag{10}$$

$$recall = \frac{TP}{TP + FN} \tag{11}$$

$$F1\ score = 2 * \frac{precision * recall}{precision + recall} \tag{12}$$

Precision is the ratio of correctly predicted positive observations of the total predicted positive observations. Recall shows the ratio of correctly predicted positive observations to all observations in the actual class. F1 score is the weighted average of Precision and Recall. Therefore, this score takes both false positives and false negatives into account.

## 4. Results

The quality of the proposed models (MultipleDgw, TfIdf, Dgw) was evaluated using evaluation measures (accuracy, precision, recall, f1-score, precision_fake, recall_fake, precision_real, recall_real). Within the k-fold validation, 10 measurements of each evaluation metric were performed for each fold.

The average values for calculated accuracies for each method and dataset are given in Table 3.

According to Table 3, better results were recorded for the Covid-19 dataset. We are presenting descriptive statistics of accuracy results for both datasets in Tables 4 and 5 for better visualization of achieved accuracies in the folds.

Table 4 shows that technique Dgw was expressively worse than TfIdf for the Covid-19 dataset. These inferior results were observed by both classification methods (Linear SVC, Random Forest). Results of the second dataset in Table 5 show, that difference between Dgw and other techniques are not so marked. Similarly, insufficient results of the Dgw technique were observed with the other evaluation measurements (precision, recall, f1-score, precision_fake, recall_fake, precision_real, recall_real), especially in the Covid-19 dataset. For this reason, we will omit this method from further evaluation of the results.

Precision results of the technique Dgw were similar to accuracy. Therefore, we visualised by the boxplot on Figs 4 and 5 only techniques MultipleDgw and TfIdf for both classification

**Table 3. Accuracy of classification (classification method / dataset).**

|  | LinearSVC | | | Random Forest | | |
| --- | --- | --- | --- | --- | --- | --- |
| Method / Dataset | MultipleDgw | TfIdf | Dgw | MultipleDgw | TfIdf | Dgw |
| Covid-19 dataset | 0.928 | 0.932 | 0.770 | 0.911 | 0.902 | 0.764 |
| KaiDMML | 0.783 | 0.788 | 0.765 | 0.792 | 0.753 | 0.773 |

**Table 4. Descriptive statistics for accuracy results for the Covid-19 dataset.**

| | Linear SVC | | | Random Forest | | |
|---|---|---|---|---|---|---|
| | **MultipleDgw** | **TfIdf** | **Dgw** | **MultipleDgw** | **TfIdf** | **Dgw** |
| Mean | 0.928 | 0.932 | 0.770 | 0.911 | 0.902 | 0.764 |
| Median | 0.926 | 0.930 | 0.761 | 0.909 | 0.900 | 0.766 |
| Standard Deviation | 0.019 | 0.022 | 0.054 | 0.031 | 0.026 | 0.052 |
| Minimum | 0.896 | 0.896 | 0.687 | 0.852 | 0.852 | 0.661 |
| Maximum | 0.966 | 0.966 | 0.845 | 0.948 | 0.940 | 0.819 |
| Lower Quartile | 0.915 | 0.915 | 0.727 | 0.899 | 0.890 | 0.752 |
| Upper Quantile | 0.940 | 0.946 | 0.819 | 0.936 | 0.920 | 0.805 |
| Skewness | 0.353 | 0.061 | 0.109 | -0.511 | -0.432 | -0.920 |
| Number of valid values | 10 | 10 | 10 | 10 | 10 | 10 |

methods. Results show that the technique MultipleDgw is better in the case of precision. Worse results were observed only for the Linear SVC method in the Covid-19 dataset. In the boxplots, we visualised the quartiles. The mean values of the results are very similar to the median values. Different results were observed for the Random Forest method in the KaiDMML dataset between the MultipleDgw (mean: 0.8632) and TfIdf (mean: 0.8061) techniques.

Precision_fake on Figs 6 and 7 and precision_real metrics on Figs 8 and 9 show worse results calculated by Linear SVC for both datasets. These worse results were observed mainly for the median values. The means of the precision results were similar in the Linear SVC method and also in precision_fake measurement, where for the Covid-19 dataset the mean of the MultipleDgw was 0.9266 and 0.9272 for the TfIdf and in the KaiDMML dataset it was 0.7622 for MultipleDgw and 0.7769 for TfIdf. In contrast to the median, the MultipleDgw technique with a mean of 0.8982 performed better compared to the TfIdf technique with a mean of 0.8913 for the Random Forest method in the Covid-19 dataset.

Precision real metric achieved better results for the MultipleDgw technique in both classification methods as well as in both datasets except for the Linear SVC method in the Covid-19 dataset. In this case, also the mean was worse.

The next used performance measurement was recall (Figs 10 and 11). The results are different concerning the used classification method in the recall metric. Worse mean and median values were observed for the MultipleDgw technique in the Linear SVC method. On the other hand, this technique achieved better results in Random Forest classification. For the

**Table 5. Descriptive statistics for accuracy results for the KaiDMML dataset.**

| | Linear SVC | | | Random Forest | | |
|---|---|---|---|---|---|---|
| | **MultipleDgw** | **TfIdf** | **Dgw** | **MultipleDgw** | **TfIdf** | **Dgw** |
| Mean | 0.783 | 0.788 | 0.765 | 0.792 | 0.753 | 0.773 |
| Median | 0.780 | 0.780 | 0.780 | 0.780 | 0.775 | 0.768 |
| Standard Deviation | 0.061 | 0.051 | 0.065 | 0.053 | 0.083 | 0.080 |
| Minimum | 0.707 | 0.707 | 0.659 | 0.725 | 0.610 | 0.650 |
| Maximum | 0.900 | 0.875 | 0.878 | 0.878 | 0.878 | 0.902 |
| Lower Quartile | 0.731 | 0.775 | 0.736 | 0.756 | 0.714 | 0.750 |
| Upper Quantile | 0.804 | 0.804 | 0.800 | 0.838 | 0.795 | 0.804 |
| Skewness | 0.715 | 0.204 | -0.220 | 0.416 | -0.567 | -0.016 |
| Number of valid values | 10 | 10 | 10 | 10 | 10 | 10 |

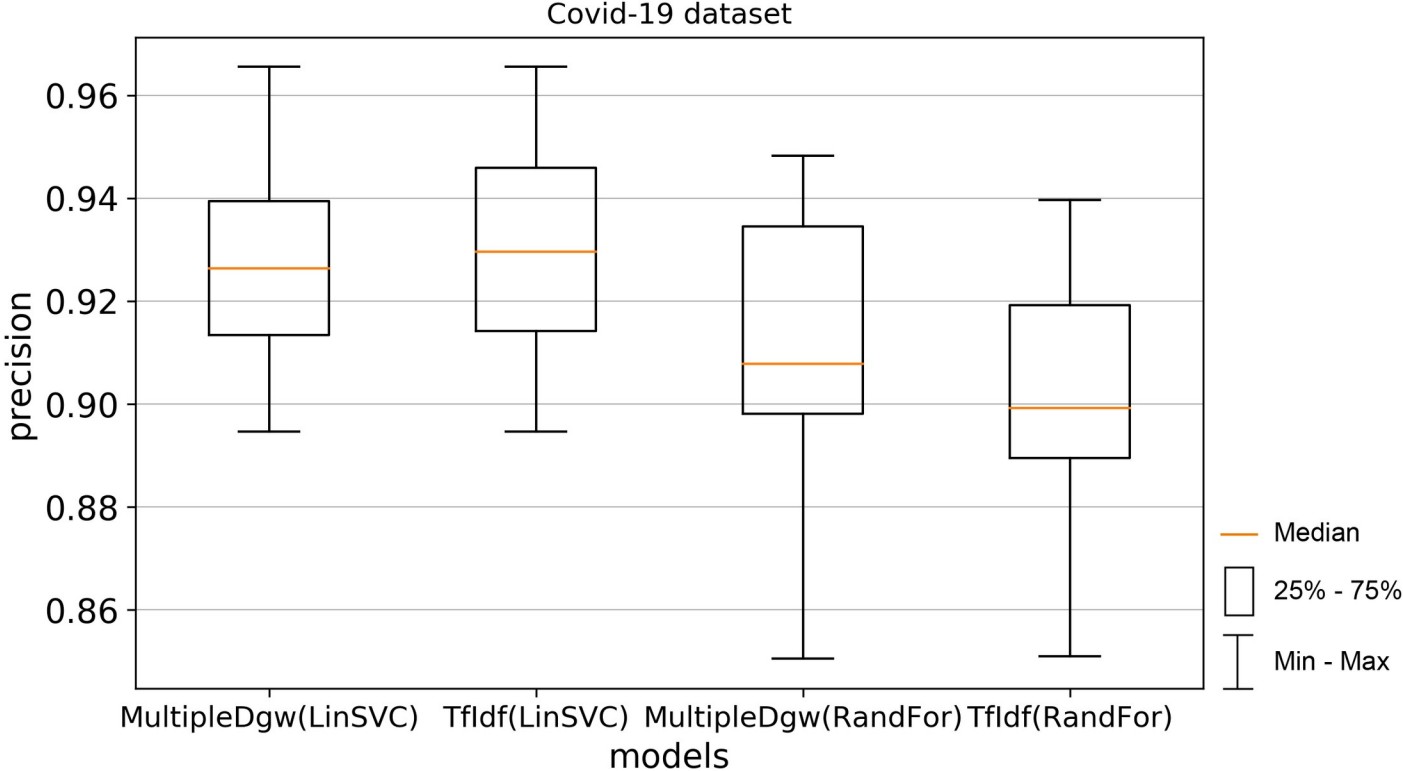

**Fig 4. Boxplot for the model performance measurements—precision for dataset Covid-19.**

KaiDMML dataset, the difference for mean in the Random Forest method was in favour of MultipleDgw with a value of 0.7823 compared to the TfIdf with a mean of 0.7489.

Finally, we calculated the f1 score (Figs 12 and 13). This score takes both false positives and false negatives into account. F1 score is usually more useful than accuracy, especially in the case of uneven class distribution. This performance measurement confirms the better results of the MultipleDgw technique compared to the TfIdf for the Random Forest classification method. Worse mean and median were observed for MultipleDgw in the Linear SVC method.

## 5. Discussion

In this paper, we focused on improving fake news classification using dependency grammar. We introduced two proprietary techniques–Dgw and MultipleDgw. The first Dgw technique achieved insufficient accuracy results. We regard accuracy as the most important performance measurement in the problem of fake news classification. For this reason, we did not provide further results for this technique. On the other hand, notable results were found in the technique MultipleDgw. However, when comparing MultipleDgw and TfIdf technique, it is clear from the descriptive statistics (Tables 4 and 5, Figs 4–13) that the differences between the methods are not statistically notable. For this reason, we did not verify the statistical significance of the differences between the methods.

The outcome showed better results for all measurements for the Covid-19 dataset. This dataset was created using existing methods of automatic news labelling. Therefore, it contains texts that have previously been evaluated by automatic methods. KaiDMML dataset was created manually, so it may not be as sensitive to automatic classifiers. In our opinion, this is the reason for its worse results.

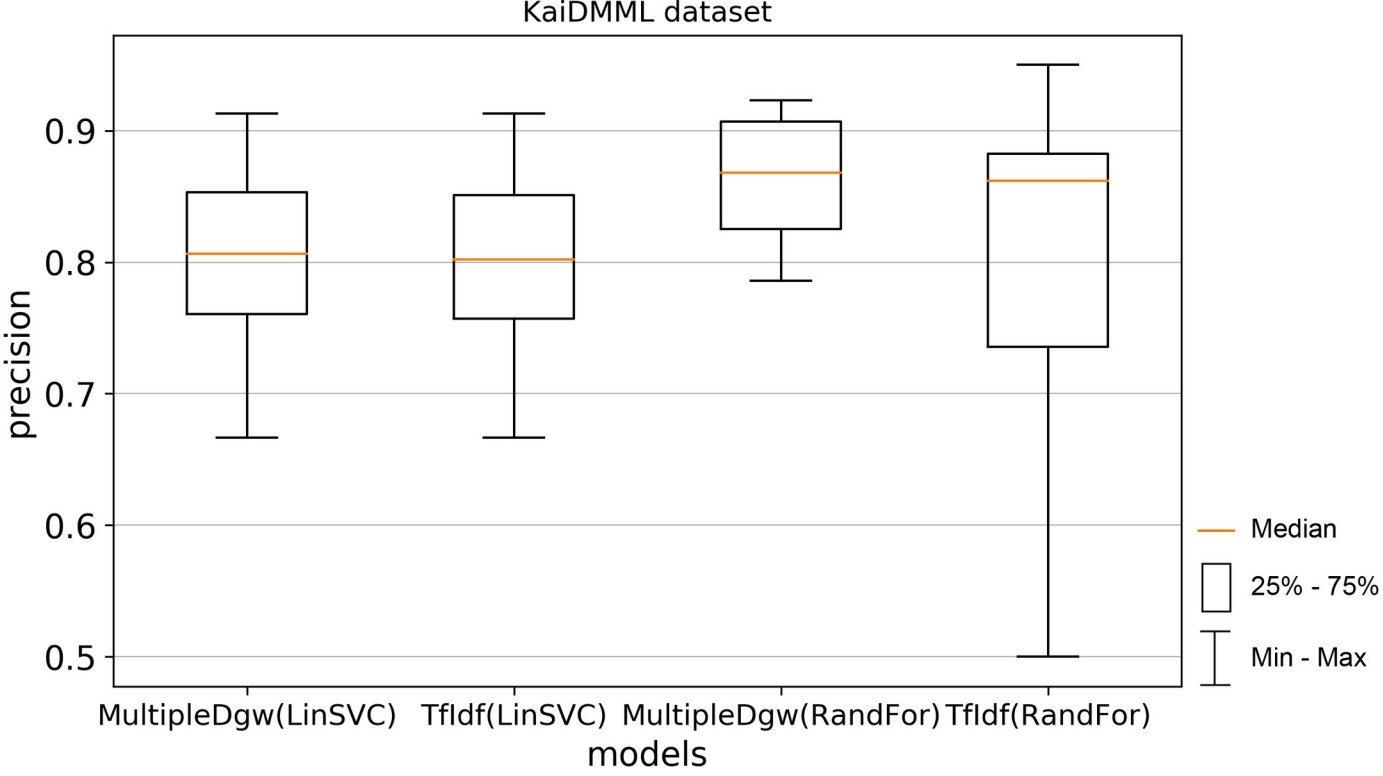

**Fig 5. Boxplot for the model performance measurements—precision for dataset KaiDMML.**

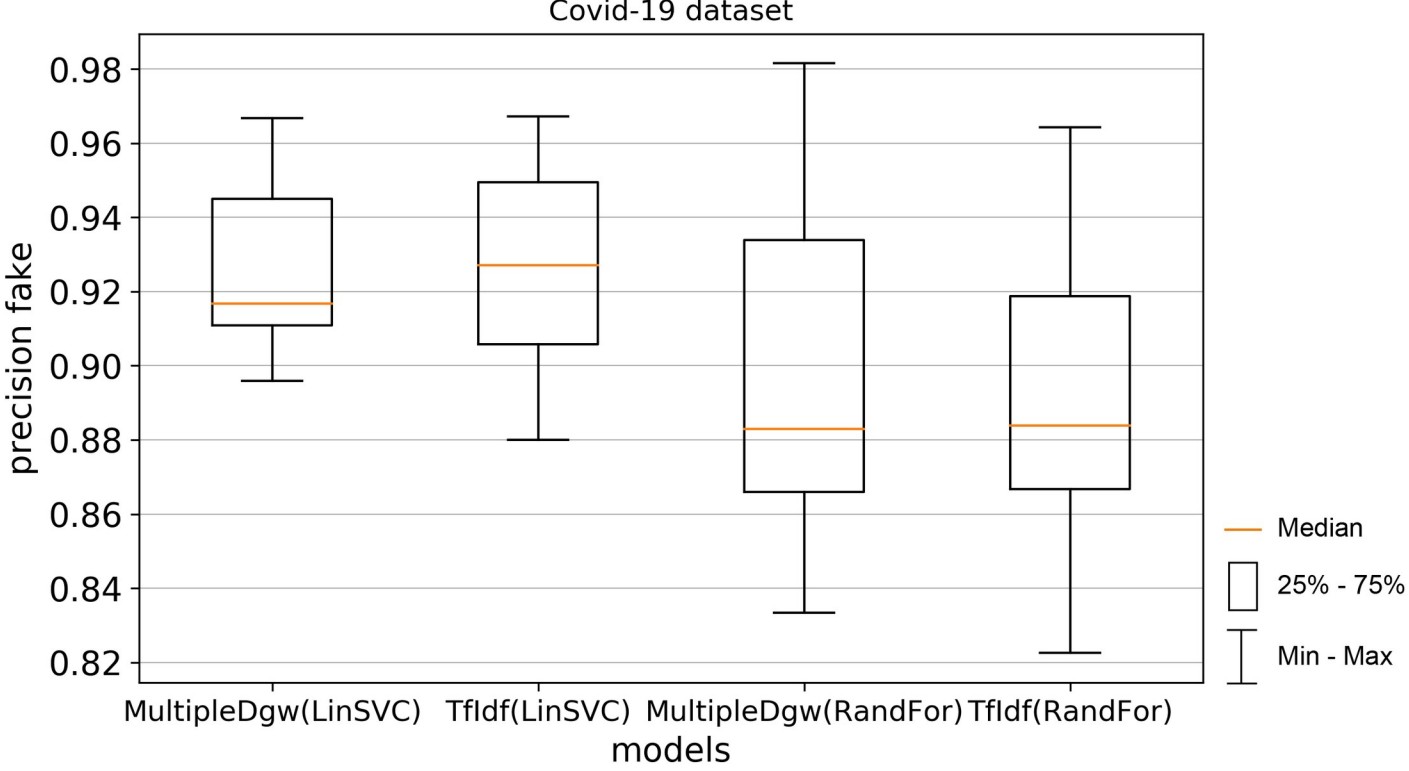

**Fig 6. Boxplot for the model performance measurement—precision_fake for dataset Covid-19.**

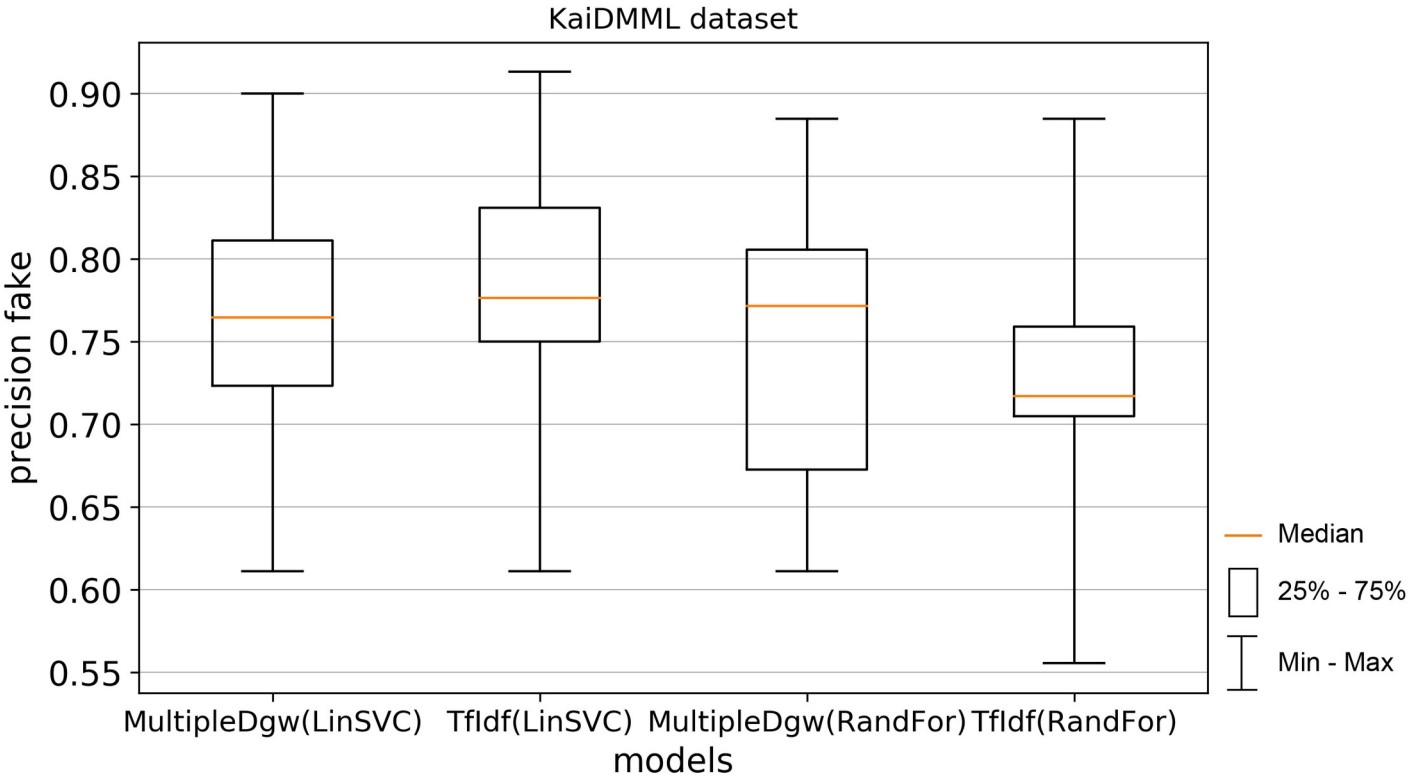

**Fig 7. Boxplot for the model performance measurement—precision_fake for dataset KaiDMML.**

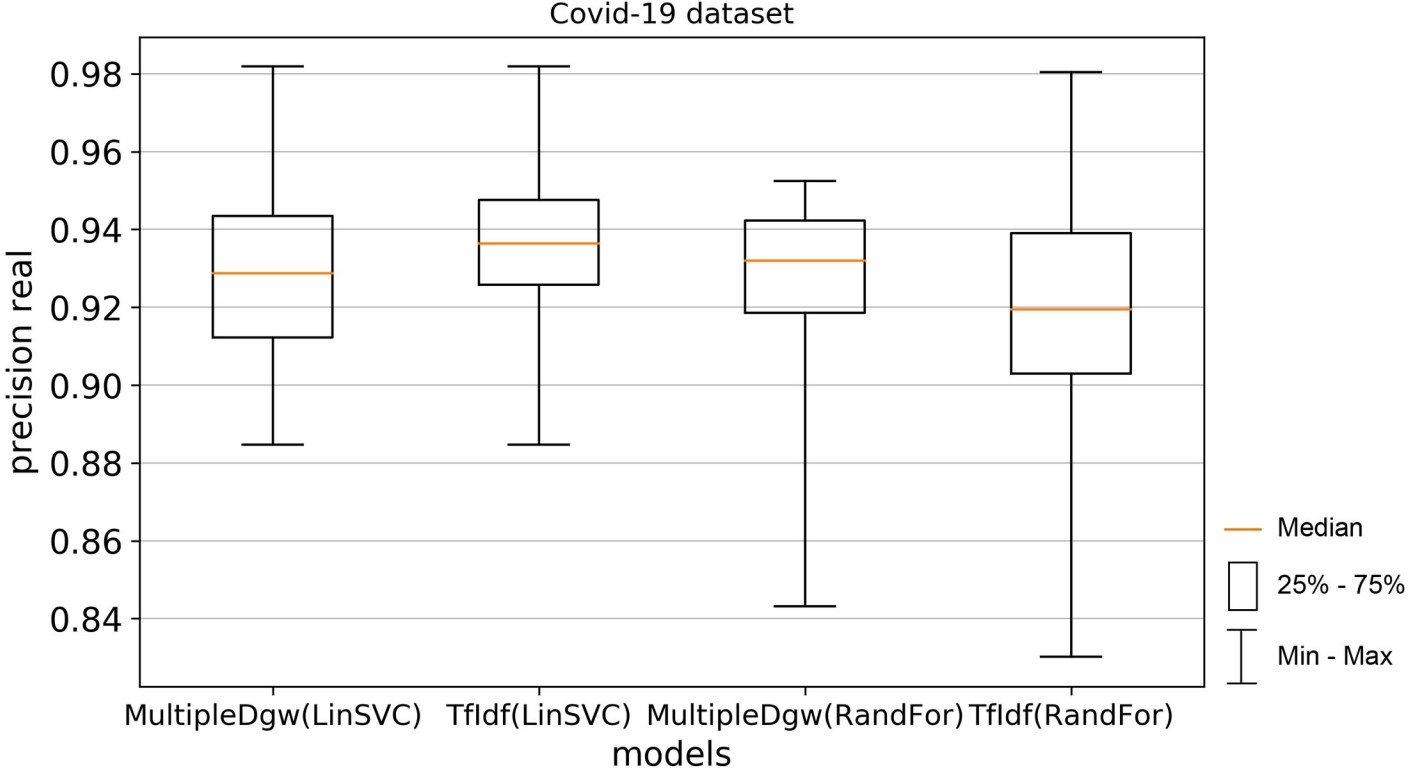

**Fig 8. Boxplot for the model performance measurement—precision_real for dataset Covid-19.**

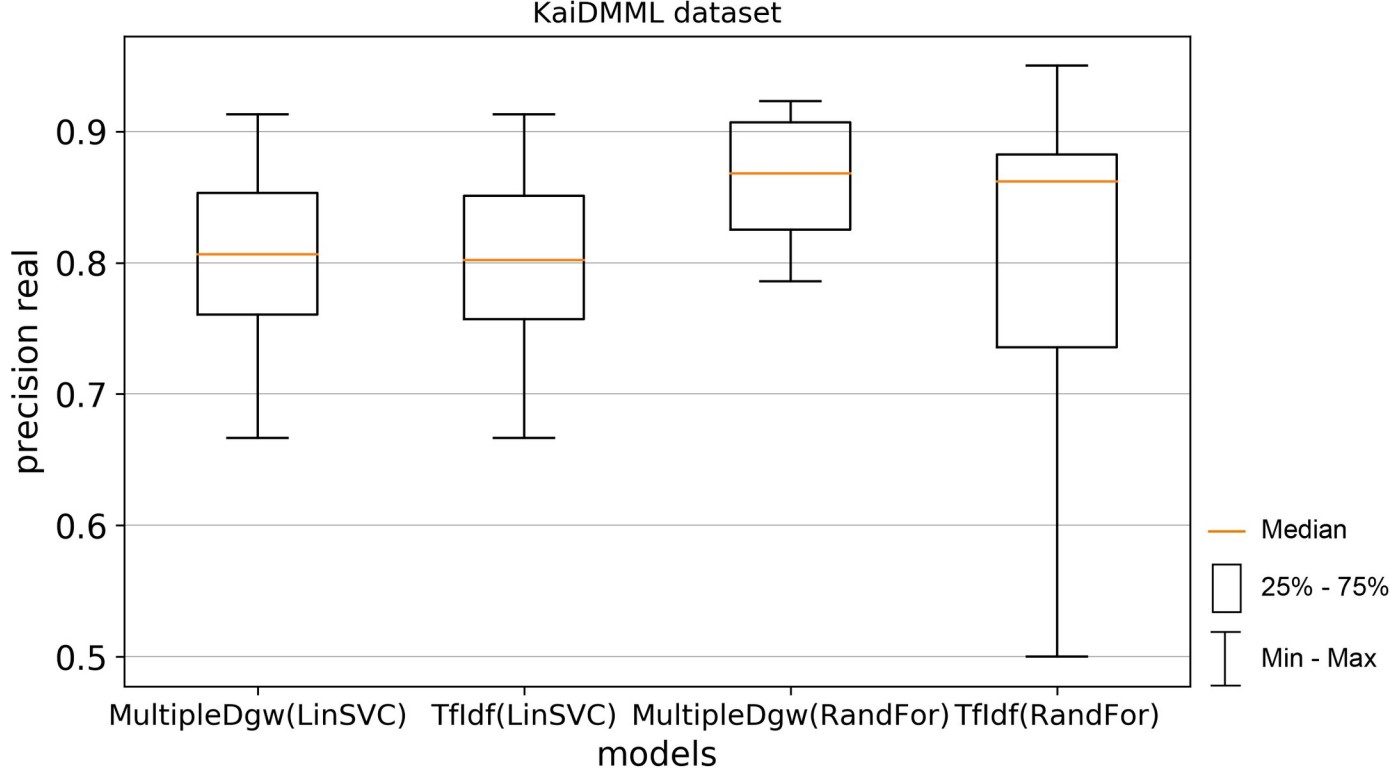

**Fig 9. Boxplot for the model performance measurement—precision_real for dataset KaiDMML.**

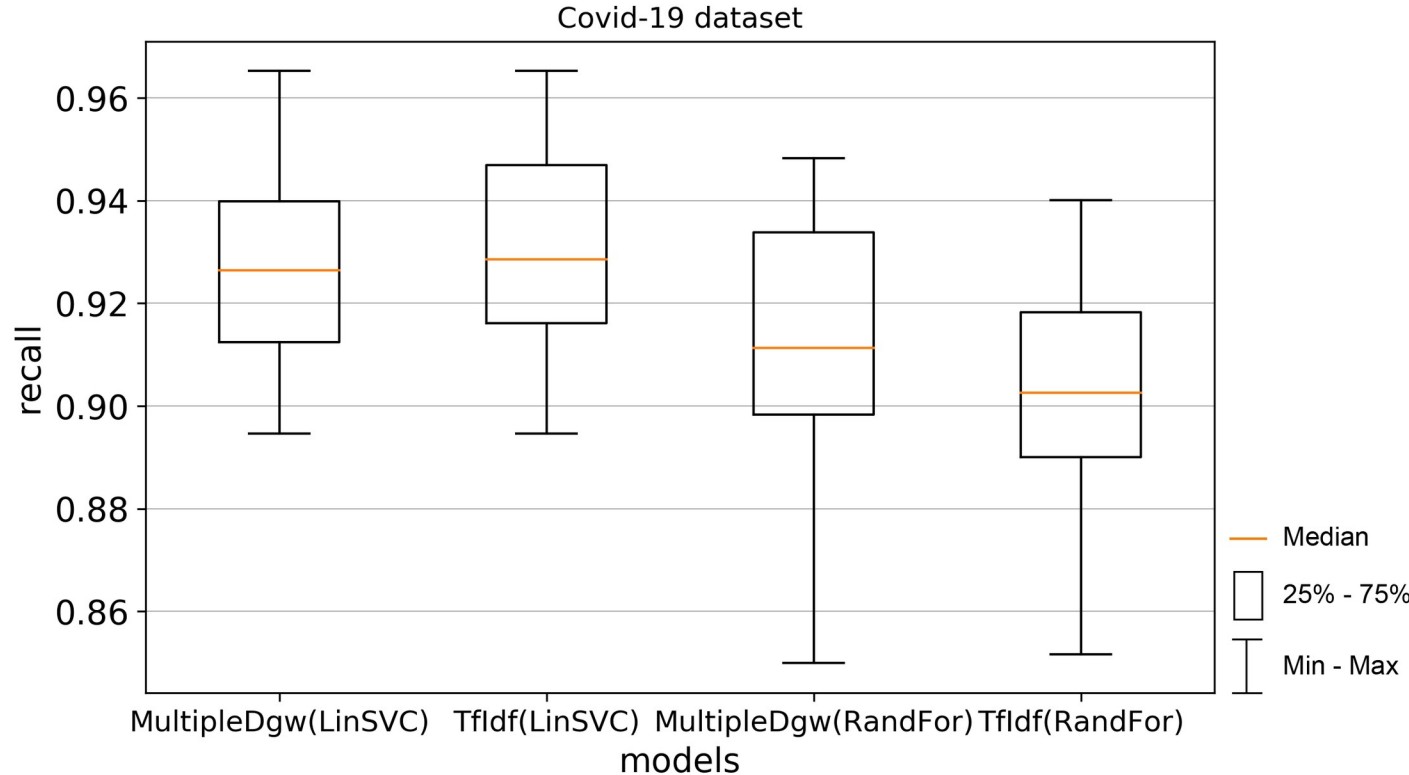

**Fig 10. Boxplot for the model performance measurement—recall for dataset Covid-19.**

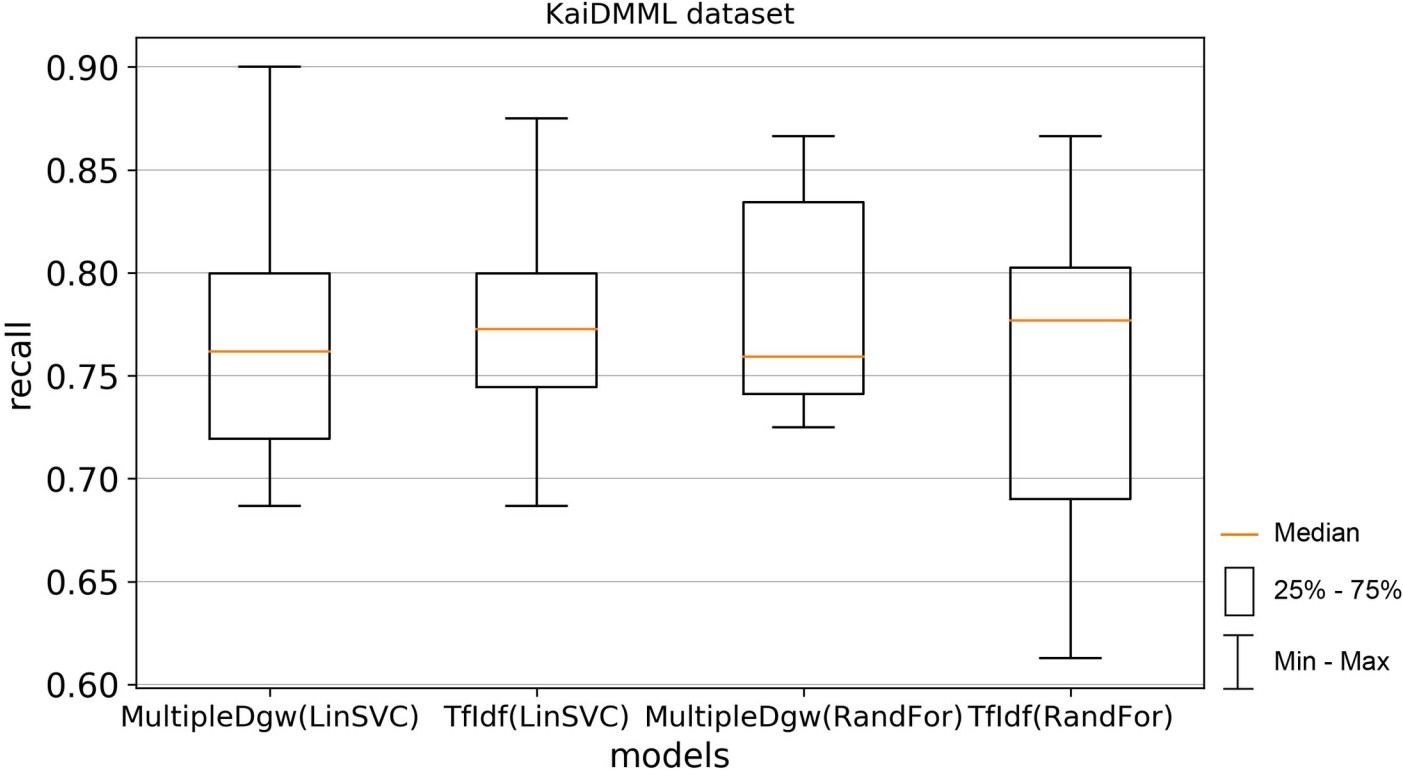

**Fig 11. Boxplot for the model performance measurement—recall for dataset KaiDMML.**

We tried to find out whether it is possible to use dependency grammar for the classification of fake news in our research. The method, based only on dependency grammar, was not successful. The creation of an input vector from the dependency grammar alone is not enough. An important result is the finding that the dependency grammar can improve the TfIdf technique. Although the results are not statistically notable, we can say from the descriptive statistics that the improved method achieved better results. Therefore, it is possible to conclude that the dependency grammar can be used to improve the classification of false news.

The results show that the MultipleDgw technique achieved better results compared to the TfIdf technique, especially when using the Random Forest classification method. The results of the MultipleDgw technique were not convincing in the second Linear SVC classification method. However, the accuracy and precision results of the MultipleDgw technique were weaker only for the Linear SVC method in the Covid-19 dataset.

We proposed two methods based on the dependency grammar. We found that only dependency grammar, in our experiment counted by the technique Dgw, is insufficient for the classification, but adding dependency grammar information to another method can improve it. It is obvious that calculations in techniques based on the dependency grammar can be several. In addition to our approach described in Chapter 3.2.2, the value of Dgw could be calculated also as $dwg(t, d) = \frac{1}{\overline{depth(t,d)}+1}$, where $\overline{depth(t, d)}$ is the mean of all values of variable *depth* for term in sentence *d*. It could also be considered to amplify the weight of *dgw(t, d)* by its square root $(dwg(t,d))^2$, or other variations of these calculations. We verified one of many approaches in our experiment.

The results of the experiment were affected by the classification methods and used datasets. We tested the suitability of our techniques on one of the newest datasets containing fake and

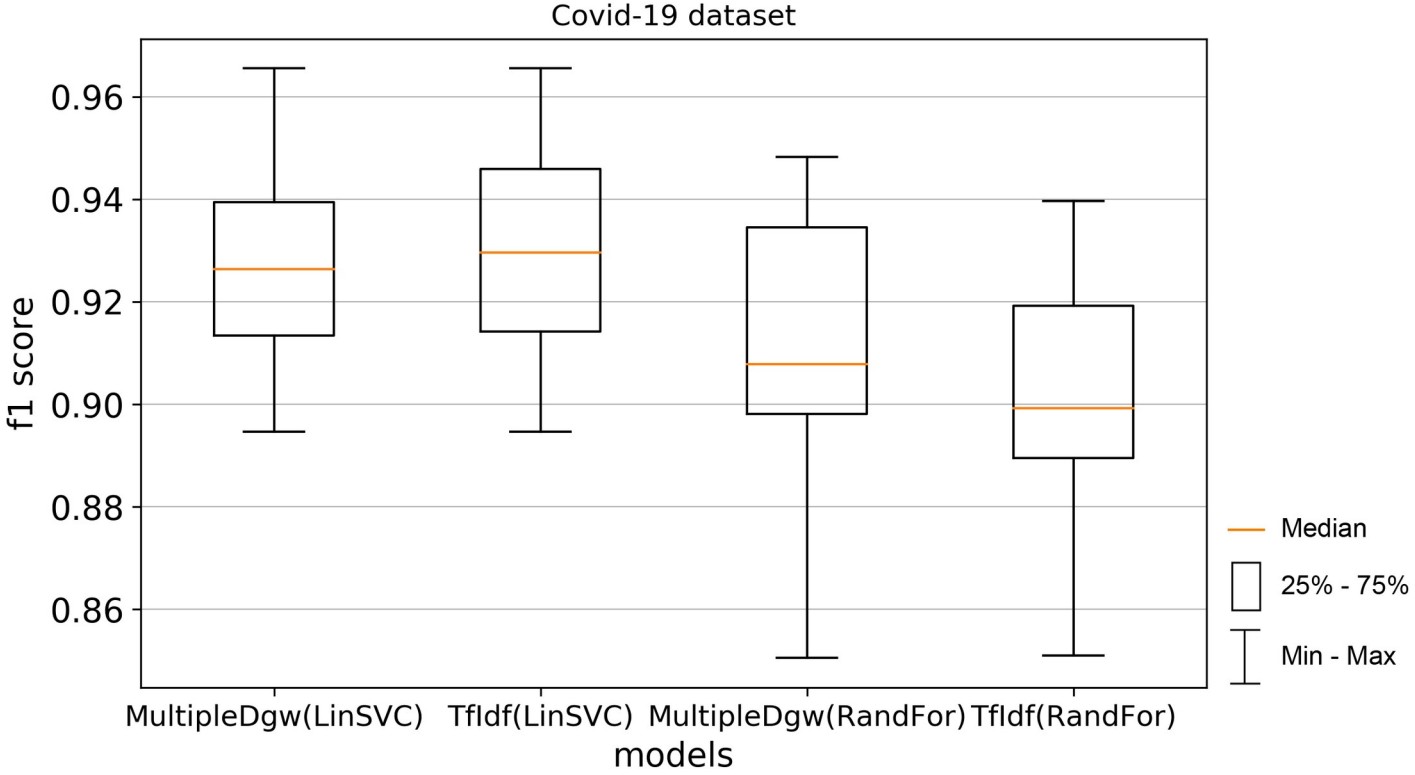

**Fig 12. Boxplot for the model performance measurement - f1-score for dataset Covid-19.**

real news about Covid-19, which contains more than 1100 reports. This dataset was created using available automatic labelling methods. The second used dataset was prepared by human evaluation.

It should be noted that in our research we used several other classification methods (Logistic Regression, SGD Classifier, Multinomial NB, Bernoulli NB, Decision Tree Classifier, Gradient Boosting Classifier). For evaluation of our proposed techniques, we chose the Linear SVC method, which of the above methods achieved the worst results to the detriment of the MultipleDgw technique and the Random Forest method with the best results for the presented technique. We did not report the results of other methods due to the scope of the article.

There are several research papers for identifying misinformation and fake news using machine learning approaches for classifications. Hakak et al. [15] used different datasets in their paper, but they chose Random Forest as one of their classification methods. They achieved worse prediction accuracy using Random Forest than us for the liar dataset, but better for the ISOT dataset. Hiramath et al. [2] used Random Forest as one of their approach with similar accuracy for fake news detection as our method with the KaiDMML dataset. Random Forest was one of the baseline methods in the paper of Sun et al. [6]. They achieved an accuracy of 0.607 on their data.

Most similar experiments focus on evaluating the performance of created models. Few works evaluate the time effectiveness of the proposed techniques. It is the time factor that appears to be limiting in the methods we propose. When compared to the traditional TfIdf technique, it is necessary to use a trained classifier to determine the word dependency and several calculations to determine the average importance of the word and these operations take time.

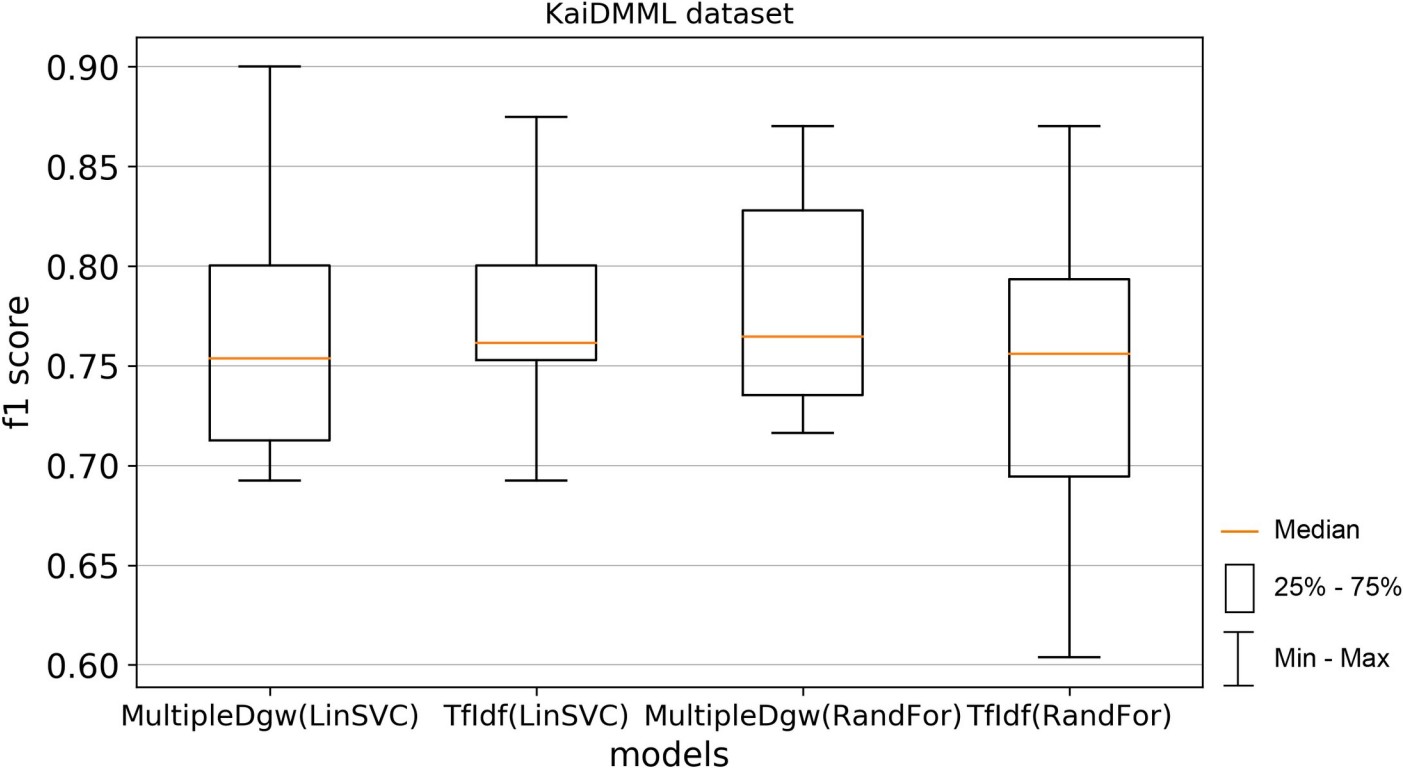

**Fig 13. Boxplot for the model performance measurement - f1-score for dataset KaiDMML.**

## 6. Conclusion

Nowadays, the classification of fake news is realised by combining several perspectives: news content, social context, credit of creator/spreader and analysing target victims. Also, the news content itself can be divided into other subgroups. A separate part is the classifiers. We focused on only a small part of this huge issue in our research. The classification methods we have chosen are used to a lesser extent in this area. It is obvious that the technique we propose will also be suitable for better classification methods.

Our goal was not to create the best classifier for fake news. By the experiments, we wanted to verify whether we can improve the classification of news using the dependency grammar. Although statistically notable differences between our two techniques were not observed, the results of the descriptive statistics showed that it is possible to improve the classification of fake news using the dependency grammar.

We first calculated the TfIdf value of the terms in the MultipleDgw technique, and these were strengthened or weakened by the importance of the word in the individual sentences. It is obvious that the MultipleDgw technique strengthened the weight of verbs, nouns, pronouns and weakened parts-of-speeches as conjunctions and prepositions.

Our proposed technique MultipleDgw achieved the best results in our experiment in the performance measurement. We have thus pointed out the correctness of our approach.

In addition to creating a suitable classification of fake news, our intention is also to understand the morphological and syntactic structures used by the creators of fake news. Decision trees or other classifiers generating understandable rules can be used as classifiers to understand the differences between fake and true news in terms of language syntax and morphology

[42]. An interesting possibility seems to be to examine n-gram sequences or POS tags using sequence analysis methods [43, 44].

We would like to focus on verifying other techniques outgoing from the dependency grammar in our future work. Some of them were presented in the discussion part of this paper. We would also like to verify our methods on neural network models and thus improve the performance of our models.

## Supporting information

**S1 File. Data for this paper.**
(XLSX)

## Author Contributions

**Investigation:** Kitti Nagy, Jozef Kapusta.

**Visualization:** Kitti Nagy.

**Writing – original draft:** Kitti Nagy, Jozef Kapusta.

**Writing – review & editing:** Kitti Nagy, Jozef Kapusta.

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
