## [Decision Letter · Decision Letter 0]

25 Jun 2021

PONE-D-21-14346

Improving Fake News Classification Using Dependency Grammar

PLOS ONE

Dear Dr. Nagyova,

Thank you for submitting your manuscript to PLOS ONE. After careful consideration, we feel that it has merit but does not fully meet PLOS ONE’s publication criteria as it currently stands. Therefore, we invite you to submit a revised version of the manuscript that addresses the points raised during the review process.

We look forward to receiving your revised manuscript.

Kind regards,

Rajesh Kaluri, Ph.D

Academic Editor

PLOS ONE

Additional Editor Comments:

Based on the comments received from the reviewers I suggest minor revisions for the paper.

Journal Requirements:

2. Please upload a copy of Figures 9-13 to which you refer in your text. If the figure is no longer to be included as part of the submission please remove all reference to it within the text.

3. Please ensure that you refer to Figures 5, 7 and 9-13 in your text and not just include the legend as, if accepted, production will need this reference to link the reader to the figure.

Reviewers' comments:

Reviewer's Responses to Questions

**Comments to the Author**

1. Is the manuscript technically sound, and do the data support the conclusions?

Reviewer #1: Yes

Reviewer #2: Yes

2. Has the statistical analysis been performed appropriately and rigorously? 

Reviewer #1: Yes

Reviewer #2: Yes

3. Have the authors made all data underlying the findings in their manuscript fully available?

Reviewer #1: No

Reviewer #2: Yes

4. Is the manuscript presented in an intelligible fashion and written in standard English?

Reviewer #1: No

Reviewer #2: Yes

5. Review Comments to the Author

Reviewer #1: Introduction needs to explain the main contributions of the work more clearly.

The Wide ranges of applications need to be addressed in Introductions

There are some grammatical and editing problems in English. English presentation should be further polished, check plagiarism.

The motivation for the present research would be clearer, by providing a more direct link between the importance of choosing your own method.

can cite the following papers like

Vehicle theft identification and intimation using gsm & iot

Constraint-based measures for DNA sequence mining using group search optimization algorithm

AN ENHANCED ALGORITHM FOR FREQUENT PATTERN MINING FROM BIOLOGICAL SEQUENCES

Reviewer #2: 1. What are the limitations of the existing works that motivated the current research?

2. List out the main contributions of the current work.

3. The recent works can be summarized in a table.

4. Some of the recent and relevant works such as "An ensemble machine learning approach through effective feature extraction to classify fake news, Propagation of Fake News on Social Media: Challenges and Opportunities, A Trusted Social Network using Hypothetical Mathematical Model and Decision-based Scheme" can be discussed in the paper.

5. Compare the current work with recent state-of-the-art.

6. Provide a detailed analysis on the results obtained.

7. The conclusion should include a brief discussion on the limitations of the current work.

6. PLOS authors have the option to publish the peer review history of their article (what does this mean?). If published, this will include your full peer review and any attached files.

Reviewer #1: No

Reviewer #2: No

---

## [Author Response · Author response to Decision Letter 0]

11 Jul 2021

Thank you for the reviewers’ effort and all suggestions. We believe that we have correctly understood the reviewers’ comments and recommendations and included them in this improved version of the manuscript. Simultaneously, we believe, we explained a possible contribution of the paper and improved its readability and comprehensibility. Besides the final decision, we appreciate the reviewers' valuable feedback, which motivated us to research fake news identification further.

---

## [Decision Letter · Decision Letter 1]

19 Aug 2021

Improving Fake News Classification Using Dependency Grammar

PONE-D-21-14346R1

Dear Dr. Nagy,

We’re pleased to inform you that your manuscript has been judged scientifically suitable for publication and will be formally accepted for publication once it meets all outstanding technical requirements.

Kind regards,

Y-h. Taguchi, Dr. Sci.

Academic Editor

PLOS ONE

Additional Editor Comments (optional):

Congratulations! Your manuscript was accepted for the publication in PLoS ONE in the present form

Thanks for submitting your valuable work to PLoS ONE.

Reviewers' comments:

Reviewer's Responses to Questions

**Comments to the Author**

1. If the authors have adequately addressed your comments raised in a previous round of review and you feel that this manuscript is now acceptable for publication, you may indicate that here to bypass the “Comments to the Author” section, enter your conflict of interest statement in the “Confidential to Editor” section, and submit your "Accept" recommendation.

Reviewer #1: All comments have been addressed

Reviewer #2: All comments have been addressed

Reviewer #3: (No Response)

2. Is the manuscript technically sound, and do the data support the conclusions?

Reviewer #1: Yes

Reviewer #2: Yes

Reviewer #3: (No Response)

3. Has the statistical analysis been performed appropriately and rigorously? 

Reviewer #1: Yes

Reviewer #2: Yes

Reviewer #3: (No Response)

4. Have the authors made all data underlying the findings in their manuscript fully available?

Reviewer #1: Yes

Reviewer #2: Yes

Reviewer #3: (No Response)

5. Is the manuscript presented in an intelligible fashion and written in standard English?

Reviewer #1: Yes

Reviewer #2: Yes

Reviewer #3: (No Response)

6. Review Comments to the Author

Reviewer #1: its difficult that to find where the comments addressed so please highlight the things where the comments addressed and then submit

Reviewer #2: The manuscript can be accepted in the present form as the authors have addressed all the comments/suggestion.

Reviewer #3: (No Response)

7. PLOS authors have the option to publish the peer review history of their article (what does this mean?). If published, this will include your full peer review and any attached files.

Reviewer #1: No

Reviewer #2: **Yes: **G Thippa Reddy

Reviewer #3: **Yes: **Rutvij Jhaveri

---

## [Editor Report · Acceptance letter]

6 Sep 2021

PONE-D-21-14346R1 

Improving fake news classification using dependency grammar 

Dear Dr. Nagy:

I'm pleased to inform you that your manuscript has been deemed suitable for publication in PLOS ONE. Congratulations! Your manuscript is now with our production department. 

Kind regards, 

on behalf of

Professor Y-h. Taguchi 

Academic Editor

PLOS ONE